# iTryOn: Mastering Interactive Video Virtual Try-On with Spatial-Semantic Guidance

**Jun Zheng** [1]   **Zhengze Xu** [2]   **Mengting Chen** [2]   **Jing Wang** [1]   **Jinsong Lan** [2]   **Xiaoyong Zhu** [2]   **Kaifu Zhang** [2]   **Bo Zheng** [2]   **Xiaodan Liang** [1]

## Abstract

Video Virtual Try-On (VVT) aims to seamlessly replace a garment on a person in a video with a new one. While existing methods have made significant strides in maintaining temporal consistency, they are predominantly confined to non-interactive scenarios where models merely showcase garments. This limitation overlooks a crucial aspect of real-world apparel presentation: active human-garment interaction. To bridge this gap, we introduce and formalize a new challenging task: Interactive Video Virtual Try-On (Interactive VVT), where subjects in the video actively engage with their clothing (e.g., pulling a hem or unzipping a jacket). This task introduces unique challenges beyond simple texture preservation, including: (1) resolving the semantic ambiguity of interactions from standard pose information, and (2) learning complex garment deformations from video where interactive moments are sparse and brief. To address these challenges, we propose **iTryOn**, a novel framework built upon a large-scale video diffusion Transformer. iTryOn pioneers a multi-level interaction injection mechanism to guide the generation of complex dynamics. At the spatial level, we introduce a garment-agnostic 3D hand prior to provide fine-grained guidance for precise hand-garment contact, effectively resolving spatial ambiguity. At the semantic level, iTryOn leverages global captions for overall context and time-stamped action captions for localized interactions, synchronized via our novel Action-aware Rotational Position Embedding (A-RoPE). Furthermore, we design an action-aware constraint loss to stabilize training and focus the learning process on these critical

interactive frames. To facilitate research and evaluation, we construct VVT-Interact, the first large-scale dataset for this task, and propose a novel interaction-aware evaluation metric to quantify the semantic fidelity of interactions. Extensive experiments demonstrate that iTryOn not only achieves state-of-the-art performance on traditional VVT benchmarks but also establishes a commanding lead in the new interactive setting, marking a significant step towards more dynamic and controllable virtual try-on experiences.

## 1. Introduction

Generative models have achieved remarkable progress, catalyzing innovations across numerous domains, with virtual try-on emerging as a quintessential application in e-commerce and digital content creation. The field initially focused on image-based virtual try-on, where early methods leveraging Generative Adversarial Networks (GANs) (Xie et al., 2021; He et al., 2022; Choi et al., 2021; Xiel et al., 2023; Xie et al., 2021) have recently been surpassed by diffusion models (Kim et al., 2024; Xu et al., 2025; Choi et al., 2024; Chong et al., 2025a), which demonstrate superior fidelity in synthesizing realistic person-garment composites. However, static images fail to capture the dynamic interplay between a garment and human motion, a crucial factor for a comprehensive apparel assessment.

Consequently, research has shifted towards the more challenging yet practical task of Video Virtual Try-On (VVT). VVT aims to generate a temporally coherent video of a person wearing a new garment, capturing its drape, flow, and response to movement. A primary obstacle that distinguishes VVT from its image-based counterpart is ensuring spatiotemporal consistency—the seamless preservation of garment texture and structure across all video frames. A naive frame-by-frame application of image try-on methods invariably leads to flickering artifacts and temporal discontinuities. To overcome this, recent VVT methods (Xu et al., 2024; Fang et al., 2024; Karras et al., 2024; Chong et al., 2025b; Li et al., 2025b; Zuo et al., 2025) have successfully

---

[1]Shenzhen Campus of Sun Yat-sen University [2]Taobao & Tmall Group of Alibaba. Correspondence to: Xiaodan Liang <xdliang328@gmail.com>.

*Proceedings of the 43rd International Conference on Machine Learning*, Seoul, South Korea. PMLR 306, 2026. Copyright 2026 by the author(s).

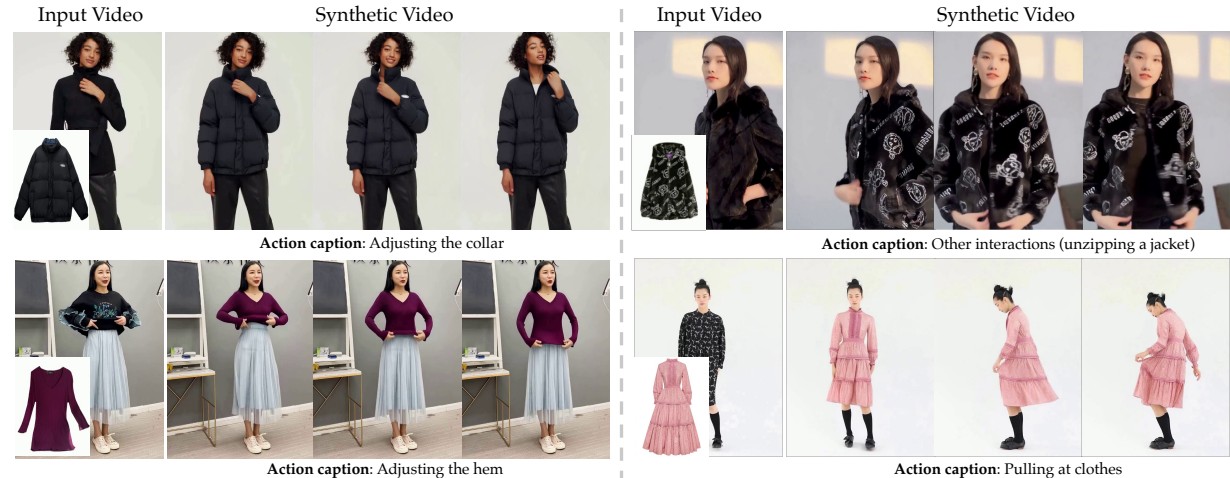

*Figure 1.* iTryOn synthesizes a diverse range of complex human-garment interactions guided by action captions. The examples showcase the model's ability to generate physically plausible deformations for various actions. (Best viewed in motion in the supplementary videos)

adapted powerful pre-trained diffusion models by incorporating temporal modules. These approaches leverage the strong priors learned from large-scale datasets to generate consistent and high-quality try-on videos, marking a significant advancement in the field. Despite this progress, existing VVT research shares a fundamental limitation: it operates exclusively within non-interactive scenarios. Current benchmarks and methods model a passive subject who simply moves or poses to display an outfit. However, the rise of live-streaming e-commerce has cultivated a new paradigm where presenters actively interact with their clothes, for example, stretching fabric to show elasticity or lifting a hem to reveal patterns. These interactions provide critical information to potential buyers but remain unaddressed by the VVT community. This discrepancy motivates us to define and tackle a new frontier: **Interactive Video Virtual Try-On (Interactive VVT)**.

The transition from non-interactive to interactive VVT introduces a unique set of challenges. The first is the semantic ambiguity of interactions. Standard conditioning signals like 2D keypoints (Yang et al., 2023) are insufficient as they lack 3D orientation and shape, making it impossible to distinguish an interactive gesture like tucking in a shirt from a non-interactive one. The second challenge is learning physical plausibility from sparse events. Interactive moments involving complex physics-driven deformations are often brief compared to simpler non-interactive segments. This imbalance creates a sparse and unstable supervisory signal, making it difficult for the model to converge on complex dynamics.

To overcome these hurdles, we propose iTryOn, a novel framework based on a large-scale video diffusion transformer that features two core innovations: a multi-level interaction injection mechanism and a targeted constraint loss. Our multi-level interaction injection mechanism resolves ambiguity by providing guidance at both spatial and semantic levels. At the spatial level, we introduce a garment-agnostic 3D hand prior to provide fine-grained guidance for the *how* of physical contact. This clean 3D reconstruction guides the model in generating accurate hand-garment contact, overcoming the limitations and information leakage of depth-based alternatives. At the semantic level, to address the *what* and *when* of an interaction, we introduce global captions for overall context and time-stamped action captions for localized control. To precisely synchronize these captions with their corresponding video segments, we design a novel Action-aware Rotational Position Embedding (A-RoPE). To address the challenge of learning from sparse events, we introduce an action-aware constraint loss. This loss function stabilizes the training process by strategically intensifying supervision on the critical but infrequent frames containing interactions. Finally, to support research and evaluation, we have curated VVT-Interact, the first large-scale dataset specifically for this task.

Our main contributions are summarized as follows: (1) We formalize the task of Interactive Video Virtual Try-On (Interactive VVT) to capture real-world human-garment interactions. To address this, we propose iTryOn, a novel framework built on a video diffusion transformer. (2) We propose a multi-level interaction injection mechanism and an action-aware constraint loss. The mechanism integrates 3D hand priors and synchronized captions to ensure precise guidance. The loss function complements this by focusing supervision on interactive frames, stabilizing the learning of complex dynamics. (3) We construct VVT-Interact, the first dataset for this task, and introduce the Interaction Success Rate (ISR) metric. Extensive experiments demonstrate that iTryOn achieves state-of-the-art performance on both

interactive and traditional benchmarks.

## 2. Related Work

### 2.1. Video Virtual Try-On

The recent proliferation of powerful open-source video generation models has catalyzed significant advancements in Video Virtual Try-On (VVT) (Xu et al., 2024; Karras et al., 2024; Fang et al., 2024; Wang et al., 2024; Li et al., 2025a; Zheng et al., 2025; Chong et al., 2025b; Li et al., 2025b; Zuo et al., 2025). Early diffusion-based methods focused on adapting image generation models for video tasks. For instance, ViViD (Fang et al., 2024) introduced a large-scale VVT dataset and repurposed an image diffusion model by inserting temporal motion modules to facilitate video-level synthesis. Subsequent works have increasingly leveraged the Diffusion Transformer (DiT) architecture, recognizing its superior capacity for spatiotemporal modeling. CatV$^2$TON (Chong et al., 2025b) proposed a unified DiT-based framework for both image and video try-on. MagicTryOn (Li et al., 2025b) built upon the powerful Wan2.1 (Wan et al., 2025) backbone, enhancing garment fidelity by injecting fine-grained guidance in the form of detailed textual descriptions and contour line maps. More recently, DreamVVT (Zuo et al., 2025) introduced a two-stage pipeline, first generating keyframes with a multi-frame try-on model and then employing another powerful video generation model to synthesize the final video from these keyframes. While these methods excel at maintaining temporal consistency for passive motion, they universally neglect active human-garment interactions. This leaves the generation of complex physics-driven interaction dynamics as a major unaddressed problem. Our work pioneers the Interactive VVT task to fill this critical gap.

### 2.2. Video Generation

Modern video generation is predominantly driven by diffusion models, with the Diffusion Transformer (DiT) architecture emerging as the state-of-the-art following the success of Sora (OpenAI, 2024). Early works like AnimateDiff (Guo et al., 2024) adapted image models with temporal modules, but recent top-performing models such as Hunyuan-DiT (Kong et al., 2025) and Wan2.1 (Wan et al., 2025) have embraced full spatiotemporal attention for superior cross-frame modeling. Our iTryOn framework builds upon this advanced lineage. We specifically adopt Wan2.1-VACE (Jiang et al., 2025) as our foundational backbone due to its strong controllable video generation capabilities. This allows us to frame video virtual try-on as a specialized video inpainting task, conditioned on a garment image for reference and human pose for structural control. Leveraging the powerful priors of Wan2.1-VACE significantly accelerates training convergence, enabling us to focus our efforts on the

novel challenges of interactive video virtual try-on.

## 3. Methodology

### 3.1. Problem Formulation

We formalize the task of Interactive Video Virtual Try-On (Interactive VVT). Given a source video $V_{\mathrm{src}} \in \mathbb{R}^{T \times 3 \times H \times W}$ depicting a person interacting with their garment, and a target garment image $G \in \mathbb{R}^{3 \times H \times W}$, the objective is to synthesize a new video $\hat{V} \in \mathbb{R}^{T \times 3 \times H \times W}$. This output video must preserve the subject's identity and motion from $V_{\mathrm{src}}$, while realistically rendering the target garment $G$ as it dynamically responds to the interaction. To achieve this, the task relies on a suite of conditional inputs $\mathcal{C}$, which includes the pose sequence $V_{\mathrm{pose}}$, a clothing-agnostic representation $V_{\mathrm{agn}}$, and specific guidance for the interaction itself. Therefore, the problem can be viewed as learning a mapping function $\mathcal{F}$ such that:

$$\hat{V} = \mathcal{F}(V_{\mathrm{src}}, G, \mathcal{C}) \tag{1}$$

Successfully learning this mapping $\mathcal{F}$ is non-trivial and introduces several unique challenges not present in traditional VVT: (1) **Interaction Ambiguity**: Standard pose skeletons are ambiguous as their 2D projection collapses motion along the Z-axis, erasing crucial depth cues. For instance, the preparatory motion of a hand moving towards the chest to button a shirt becomes nearly invisible in 2D, depriving the model of the key "approaching" signal needed to anticipate contact and thus necessitating richer 3D guidance. (2) **Learning Physical Plausibility from Sparse Events**: While the ultimate goal is to generate physically plausible dynamics, learning this from video data presents a significant challenge. Interactive moments involving complex deformations are often brief and infrequent compared to simpler, non-interactive segments. This imbalance creates a sparse and unstable supervisory signal, where the gradient from easier, static frames can overwhelm the crucial but rare signal from interactive frames. Consequently, the model may fail to converge on complex dynamics, defaulting to simpler, non-interactive generations. (3) **Data and Evaluation Scarcity**: A significant bottleneck is the lack of resources. Existing VVT datasets consist almost entirely of non-interactive sequences. Furthermore, standard metrics focus on visual fidelity but fail to verify if the human-garment interaction was semantically successful. This absence of data and specialized metrics hinders the development and benchmarking of interactive models.

To address these challenges, we adopt a comprehensive approach. First, we construct a new large-scale dataset with detailed annotations designed to resolve ambiguity. Second, we propose the iTryOn framework, an architecture designed to generate physically plausible results based on this data. Finally, we introduce the Interaction Success Rate

(ISR) metric to establish a rigorous standard for quantifying interaction fidelity in this new task.

## 3.2. Data Collection and Annotation of VVT-Interact

### 3.2.1. DATA SOURCING AND FILTERING

We initiated the process by extensively collecting video-garment pairs from e-commerce live streams and social media, which serve as rich sources for interactive clothing demonstrations. Recognizing the noisy nature of this raw data, we implemented a rigorous, multi-stage curation pipeline to ensure high quality and relevance. The pipeline first filters out unqualified data by: (1) removing pairs with low-resolution garment images; (2) discarding videos with low bitrates or significant visual artifacts; (3) excluding videos where the person occupies a small screen ratio; (4) eliminating instances where the garment is subject to unrecoverable occlusion; and (5) removing videos with scene cuts to ensure temporal continuity, using an automatic shot detection algorithm (Soucek & Lokoc, 2024).

### 3.2.2. VLM-BASED ANNOTATION FOR SEMANTIC GUIDANCE

The cornerstone of our dataset is its detailed annotation of interactions, designed to provide the multi-level semantic guidance required to resolve the interaction ambiguity challenge. We leveraged the advanced capabilities of Qwen-VL (Bai et al., 2025) to generate two distinct types of annotations: global captions and time-stamped action captions. Our annotation strategy proceeded as follows: (1) Global Caption Generation: We first prompted Qwen-VL to produce a high-level summary of the overall human motion in each video. This resulting global caption provides general context for the entire sequence. (2) Time-stamped Action Caption Generation: To pinpoint the exact temporal boundaries of interactions, we performed a fine-grained analysis. This involved tasking Qwen-VL to classify each frame as either "interactive" or "non-interactive" based on a sequence of input frames, yielding binary labels. As the initial sequence of labels was often noisy, we applied morphological smoothing to denoise the predictions and identify continuous interaction segments. Finally, we combined these temporal boundaries with a pre-determined interaction category to automatically generate the time-stamped action captions, structured as ("action description", [start_frame, end_frame]).

The final VVT-Interact dataset consists of 5,292 high-quality video-garment pairs, covering six distinct interaction categories, each annotated with both a global caption and one or more time-stamped action captions. Crucially, these precise annotations not only supervise the model training but also serve as the ground truth for our proposed Interaction Success Rate (ISR) evaluation metric. We provide a comprehensive breakdown of our data annotation pipeline in **Appendix A.2**.

## 3.3. Overview of the iTryOn Framework

The overall architecture of our proposed framework, iTryOn, is depicted in Figure 2. Built upon a conditional Diffusion Transformer (DiT) backbone, iTryOn is specifically designed to address the challenges outlined in our problem formulation. It processes a source video, a target garment, and a suite of conditional inputs to generate a realistic interactive try-on video. Guidance is injected into the DiT backbone through a set of parallel trainable modules. These include Context Blocks that process general body information (from pose and agnostic inputs) to ensure proper overall garment alignment, and our novel Interaction Guider which handles the fine-grained hand-garment contact. For efficiency, all guidance modules adopt a streamlined shared architecture, and we use only $\frac{N}{2}$ Context Blocks. The framework's core innovations are three-fold, each corresponding to a subsequent section: (1) A **fine-grained spatial guidance** mechanism processes 3D hand representations to control the precise physical contact in an interaction (Sec. 3.4). (2) An **action-aware semantic guidance** mechanism leverages time-stamped captions and our Action-aware Rotational Position Embedding (A-RoPE) to interpret the *what* and *when* of an interaction (Sec. 3.5). (3) An **action-aware constraint loss** is used during training to stabilize learning from sparse interactive events, focusing the model on complex dynamics to improve physical plausibility (Sec. 3.6).

The general data flow involves encoding all inputs into the latent space using a frozen Wan encoder, followed by an iterative denoising process within the DiT where our guidance is injected. The final denoised latents are then decoded back into the output video. The following sections will elaborate on each of these key components.

## 3.4. Fine-grained Spatial Guidance for Hand-Garment Interaction

Accurately modeling the *how* of an interaction requires resolving the spatial ambiguity inherent in 2D pose estimations (DWPose (Yang et al., 2023), DensePose (Güler et al., 2018)). This ambiguity is twofold: 2D projections lack hand shape, making it impossible to distinguish a pulling pinch from a pressing flat palm, and they lack hand orientation, failing to differentiate an interactive gesture from a non-interactive one. To address this fundamental limitation, we introduce a fine-grained spatial guidance mechanism. The choice of the geometric prior for this mechanism is critical. As illustrated in Figure 3, alternatives like hand depth are also flawed, suffering from information leakage that contaminates the conditioning signal.

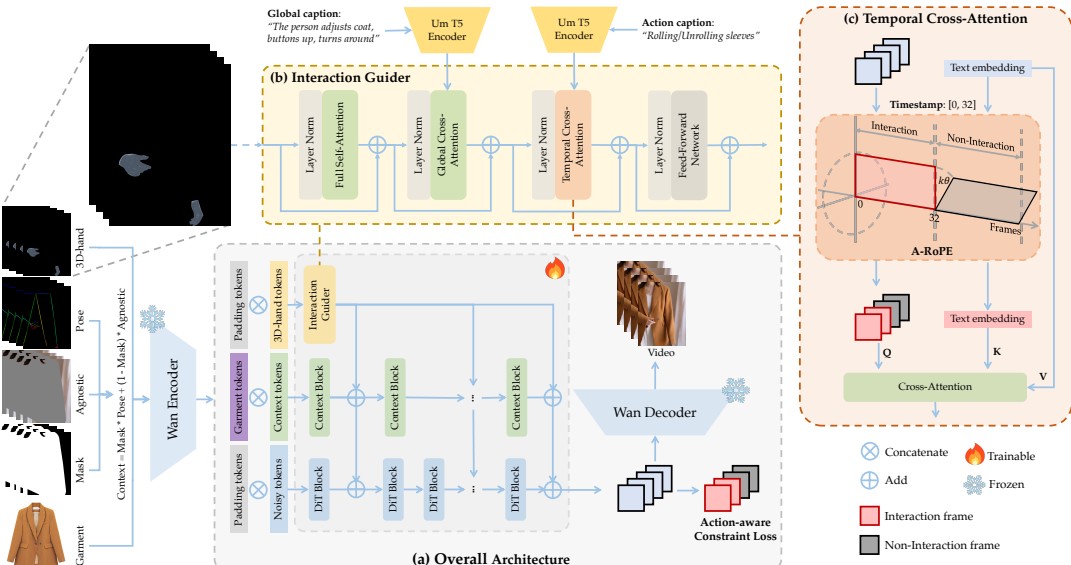

*Figure 2.* The iTryOn architecture. (a) A DiT backbone with parallel injection of general context and 3D-hand guidance from our Interaction Guider. An action-aware constraint loss focuses training on interaction frames. (b) The Interaction Guider module fuses spatial features with global and action-specific text prompts. (c) Our A-RoPE mechanism aligns action captions to their corresponding video segments via unique rotational position encodings in temporal cross-attention.

In contrast, we select a 3D hand representation as our prior, which is both detailed and garment-agnostic. We leverage the HaMeR model (Pavlakos et al., 2024) to extract this 3D hand prior, denoted as $V_{\text{hand}} \in \mathcal{C}$. As depicted in Figure 2(a), this clean geometric signal is processed by a lightweight Interaction Guider module. Concurrently, broader contextual information from the pose $V_{\text{pose}}$ and agnostic video $V_{\text{agn}}$ is handled by parallel Context Blocks. The features from both the Interaction Guider and Context Blocks are then additively fused with the video tokens at each block of the DiT backbone. This injection of precise 3D hand geometry provides the model with explicit cues about hand shape, orientation, and proximity, guiding it to generate physically plausible and accurate hand-garment contact.

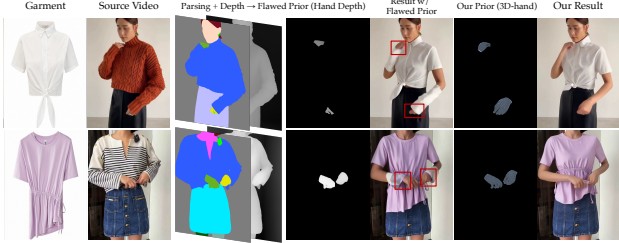

*Figure 3.* Visual justification for our garment-agnostic 3D hand prior. Deriving a "Hand Depth" prior from human parsing (Li et al., 2022) and video depth (Chen et al., 2025) suffers from critical information leakage. This flawed prior improperly retains source garment geometry, such as the sleeve cuff, leading directly to visible artifacts in the generated output. In contrast, our fully garment-agnostic 3D hand prior provides a clean signal, enabling the generation of plausible and artifact-free hand-garment contact. See **Appendix A.3** for more details.

## 3.5. Action-aware Semantic Guidance

While our spatial guidance resolves the *how* of an interaction, ambiguity remains concerning the *what* (the type of action) and the *when* (its precise timing). Although the global caption provides a high-level summary of the overall motion, we observed that its descriptions are often too generic to guide specific interactions (see **Appendix A.4.1** for detailed examples). This semantic ambiguity necessitates a more explicit form of guidance.

To address this, we introduce Action-aware Semantic Guidance, a mechanism composed of two key components: action captions for semantic specificity and an Action-aware Rotational Position Embedding (A-RoPE) for temporal precision. First, to specify the *what*, we complement the global caption with a categorical action caption drawn from a pre-defined set of interaction types. This provides the model with an unambiguous fine-grained signal about the intended action. However, interactions typically occur only within a short segment of the full video clip. Simply injecting this action caption via standard cross-attention can lead to temporal misalignment, where the semantic guidance "bleeds" into non-interactive frames. To enforce precise synchronization and control the *when*, we design A-RoPE, a novel embedding strategy inspired by MinT (Wu et al., 2025). As conceptualized in Figure 2(c), A-RoPE applies a scaled 1D-RoPE (Su et al., 2024) to distinguish between interactive and non-interactive segments based on their segment index $i$:

$$\hat{Q}_i = \text{A-RoPE}(Q_i, i) = \text{1D-RoPE}(Q_i, i \cdot k)$$
$$\hat{K}_i = \text{A-RoPE}(K_i, i) = \text{1D-RoPE}(K_i, i \cdot k)$$

(2)

where $k$ is a hyperparameter controlling the separation scale, which we set to 4 in our experiments (see Table 7 for ablation results). While A-RoPE is applied to the queries $Q_i$ of all video segments to preserve their global temporal order, it is applied to the keys $K_i$ only when they correspond to meaningful action captions from interactive segments. For non-interactive segments, we use a null caption, such as an empty string, and the resulting keys do not receive A-RoPE encoding. The value sequence $V$ is derived from the action caption embeddings without any positional encoding. The final temporal cross-attention is computed as Attention$(\hat{Q}, \hat{K}, V)$. This design ensures that the temporally scaled positional signal is activated exclusively for genuine interactions, effectively creating a dedicated temporal channel for each action-video pairing. By aligning the positional encodings of a video segment's query $\hat{Q}_i$ with those of its corresponding action caption's key $\hat{K}_i$, the attention mechanism is strongly biased toward the correct text-video alignment. This synchronization provides semantic guidance with high temporal fidelity, enabling the model to generate interaction motions that are accurate in both semantics and timing. See **Appendix A.4** for more details.

### 3.6. Action-aware Constraint Loss

To address the challenge of learning from sparse interactive events, we introduce an action-aware constraint loss (AC loss). Our guidance mechanisms provide the model with the necessary cues, but the inherent imbalance between frequent non-interactive frames and rare interactive frames can lead to training instability. The sparse gradient from complex deformations can be overwhelmed by the dense gradient from simpler frames, causing the model to neglect the crucial interaction dynamics. The AC loss counteracts this by amplifying the supervisory signal specifically on frames where interactions occur. The core idea is to strategically re-weight the standard diffusion loss, compelling the model to prioritize these critical moments. We leverage the temporal boundaries from our action captions to construct a binary mask $\mathbb{M}_{\text{action}}$ which is set to 1 for frames within an interaction segment and 0 otherwise. The overall training objective is formulated as:

$$
\begin{aligned}
\mathcal{L} = \mathbb{E}_{t,\mathbf{z}_t,c,v\sim\mathcal{N}(0,\mathbf{I})} & \left[ \| v_\theta\left(\mathbf{z}_t, t, c\right) - v \|_2^2 \right] \\
+ \lambda \mathbb{E}_{t,\mathbf{z}_t,c,v\sim\mathcal{N}(0,\mathbf{I})} & \left[ \| \mathbb{M}_{\text{action}} \odot \left( v_\theta\left(\mathbf{z}_t, t, c\right) - v \right) \|_2^2 \right],
\end{aligned}
\tag{3}
$$

where $z_t$ is the noisy latent at timestep $t$, $c$ represents the conditioning information, and $v_\theta(\cdot)$ is the v-prediction network. The first term is the standard diffusion loss computed over all frames. The second term weighted by a hyperparameter $\lambda$ (set to 0.5 in our experiments, see Table 8 for ablation results) applies an additional penalty exclusively to the latent features corresponding to the interaction frames,

as selected by the element-wise multiplication with the mask $\mathbb{M}_{\text{action}}$. By applying this targeted supervisory signal, we prevent the model from ignoring the sparse but vital interaction dynamics. This focused training approach accelerates convergence on complex motions and significantly increases the success rate of generating the intended interaction, ultimately leading to more physically plausible results.

## 4. Experiments

### 4.1. Datasets and Metrics

**Datasets.** We conduct a comprehensive evaluation of our method on both traditional non-interactive and our newly proposed interactive video virtual try-on tasks. For the non-interactive VVT task, we benchmark our model on the widely-used ViViD dataset (Fang et al., 2024). The dataset comprises 7,759 paired videos for training and 180 videos for testing, all at a resolution of 624×832. To evaluate performance on our proposed interactive VVT task, we introduce the VVT-Interact dataset. Our dataset consists of 5,160 videos for training and 132 videos for testing. To ensure a fair and robust comparison against the non-interactive benchmark, the test set was curated to have a total of 10,692 frames, which is comparable to the 11,700 total test frames in the ViViD benchmark.

**Evaluation Metrics.** To assess the performance of our method, we employ a comprehensive set of metrics divided into two categories: (1) Visual Fidelity Metrics: We use Structural Similarity (SSIM) (Wang et al., 2004) and Learned Perceptual Image Patch Similarity (LPIPS) (Zhang et al., 2018) to measure spatial reconstruction quality. Video Fréchet Inception Distance (VFID) (Dong et al., 2019) is used to assess spatiotemporal feature quality. (2) Interaction Fidelity Metrics: Standard metrics are often "blind" to the semantic success of an interaction. To address this, we use Fréchet Video Distance (FVD) (Unterthiner et al., 2019) to evaluate the temporal coherence and realism of the motion. Furthermore, we propose a novel semantic metric, the Interaction Success Rate (ISR).

**Interaction Success Rate (ISR).** ISR leverages a Vision-Language Model (VLM) to semantically "ground" the generated action. Specifically, for each test sequence, we first map the ground truth interaction segment. Then, we employ Qwen-VL (Bai et al., 2025) to perform a binary verification on the generated frames, determining if the intended interaction (e.g., "zipping up") is semantically recognizable and coherent with the hand motion. Let $N$ be the total number of interactive frames and $X$ be the number of successfully detected frames, ISR is calculated as: ISR $= \frac{X}{N}$. This metric provides a direct measure of the model's ability to generate human-garment interaction.

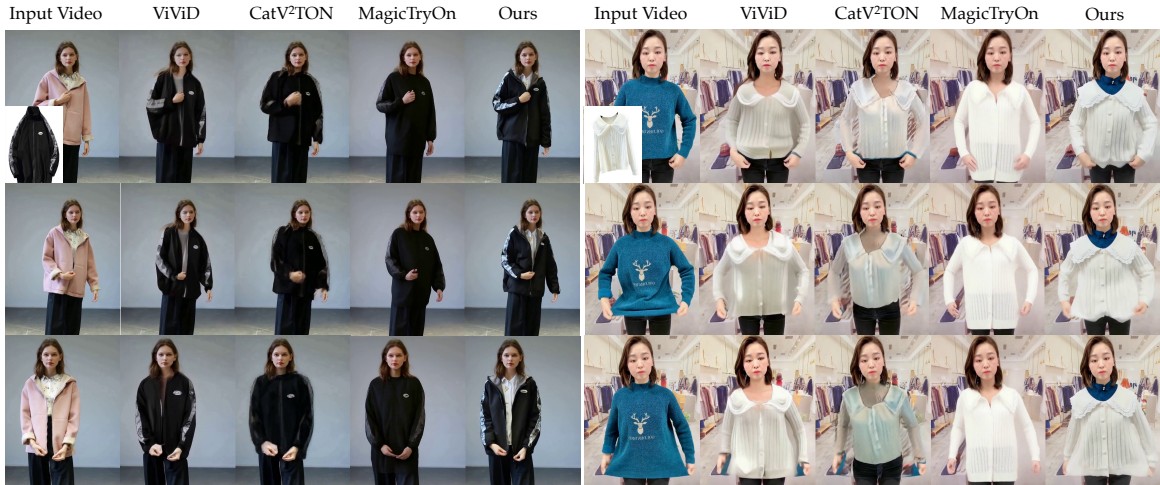

*Figure 4.* Qualitative comparison on the VVT-Interact dataset.

## 4.2. Implementation Details

Our model is initialized from the pre-trained Wan2.1-VACE (Jiang et al., 2025) and trained using a two-stage scheme. In the first stage, we finetune the model on the ViViD dataset for 10k steps using empty action captions (i.e., treating all samples as non-interactive). After this stage, we evaluate the model on the ViViD-S-Test (Chong et al., 2025b) to ensure a fair comparison with existing methods that are trained exclusively on ViViD. In the second stage, we continue training on our VVT-Interact dataset for an additional 10k steps to incorporate interactive capabilities. Throughout training, we use 81-frame video clips at a resolution of 576×768 with a per-GPU batch size of 1. We employ the AdamW optimizer (Loshchilov & Hutter, 2018) with a learning rate of 1e-5. All experiments were conducted on 8 NVIDIA A100 (80GB) GPUs. For inference, we use 50 denoising steps and a CFG scale of 3.

*Table 1.* Quantitative comparison of *Visual Fidelity* on the VVT-Interact dataset. $p$ and $u$ denote the paired and unpaired settings, respectively.

| Method | $VFID_I^p \downarrow$ | $VFID_R^p \downarrow$ | SSIM↑ | LPIPS↓ | $VFID_I^u \downarrow$ | $VFID_R^u \downarrow$ |
|---|---|---|---|---|---|---|
| ViViD (Fang et al., 2024) | 29.8272 | 1.2735 | 0.7259 | 0.1637 | 36.5179 | 1.6128 |
| CatV²TON (Chong et al., 2025b) | 26.9919 | 2.2692 | 0.7761 | 0.1434 | 36.4519 | 2.6764 |
| MagicTryOn (Li et al., 2025b) | 27.6716 | 2.6022 | 0.7649 | 0.1702 | 36.0322 | 3.3669 |
| iTryOn (ours) | **22.4640** | **0.6033** | **0.7849** | **0.1217** | **35.0479** | **1.2378** |

*Table 2.* Quantitative comparison of *Interaction Fidelity* on the VVT-Interact dataset.

| Method | $FVD^p \downarrow$ | $ISR^p \uparrow$ | $FVD^u \downarrow$ | $ISR^u \uparrow$ |
|---|---|---|---|---|
| ViViD (Fang et al., 2024) | 468.4750 | 0.3968 | 482.2153 | 0.3888 |
| CatV²TON (Chong et al., 2025b) | 533.2168 | 0.4838 | 542.4718 | 0.4245 |
| MagicTryOn (Li et al., 2025b) | 431.7865 | 0.4348 | 432.3735 | 0.4474 |
| iTryOn (ours) | **380.5578** | **0.6100** | **393.0552** | **0.6147** |

## 4.3. Quantitative Results

We quantitatively evaluate iTryOn against state-of-the-art methods on the VVT-Interact dataset. The results are presented in Table 1 and Table 2, categorizing performance into visual fidelity and interaction fidelity.

**Visual Fidelity.** As shown in Table 1, iTryOn significantly outperforms all baselines in spatial quality metrics (SSIM, LPIPS) and spatiotemporal feature consistency (VFID). This indicates that our model, despite focusing on complex interactions, maintains superior garment texture details and reduces flickering artifacts.

**Interaction Fidelity.** Table 2 highlights the decisive advantage of our method in generating realistic and semantically correct interactions. In terms of temporal coherence (FVD), iTryOn achieves the lowest score, reflecting smoother and more natural motion dynamics. Crucially, on our proposed ISR metric, iTryOn establishes a commanding lead, achieving success rates of over 61% compared to less than 49% for existing methods. This quantitative gap confirms that while baseline models may generate visually plausible frames, they often fail to execute the specific physical interaction.

*Note:* We also evaluated our model on the traditional non-interactive ViViD benchmark. iTryOn achieves state-of-the-art performance in this setting as well. Due to space constraints, detailed results and visualizations for ViViD are provided in **Appendix A.6**.

## 4.4. Qualitative Results

We provide qualitative comparisons in Figure 4 to visually substantiate our quantitative dominance in the interactive setting. Figure 4 illustrates the failure of existing methods on our VVT-Interact dataset. When faced with a zippering motion, baseline approaches either generate physically

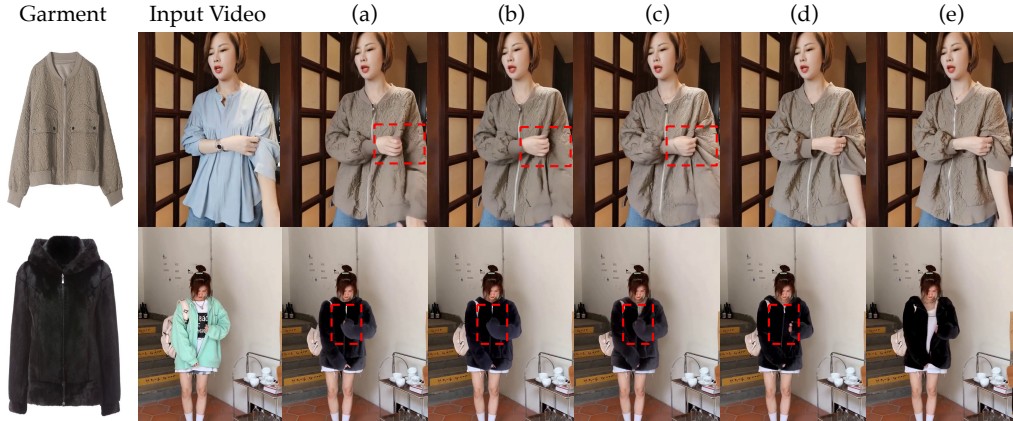

*Figure 5.* Visual comparison of different variants on the VVT-Interact dataset.

implausible deformations (e.g., ViViD) or completely misinterpret the action, producing a simple hand-gliding motion without engaging the garment (e.g., CatV$^2$TON, Magic-TryOn). Similarly, for a hem-pulling action, they often render a static unresponsive garment. In contrast, iTryOn is the only method that successfully synthesizes these interactions with high physical realism, accurately depicting the fabric zippering and stretching in response to actions. These results powerfully demonstrate the unique capability of our framework to render complex dynamic interactions.

*Table 3.* Ablation study of *Visual Fidelity* on the VVT-Interact dataset.

| ID | Method | VFID$_I^p$ ↓ | VFID$_R^p$ ↓ | SSIM↑ | LPIPS↓ | VFID$_I^u$ ↓ | VFID$_R^u$ ↓ |
|---|---|---|---|---|---|---|---|
| (a) | Baseline | 27.1203 | 1.3242 | 0.7720 | 0.1670 | 36.8623 | 1.5241 |
| (b) | (a) + Data | 26.6500 | 0.8028 | 0.7759 | 0.1338 | 36.5853 | 1.3677 |
| (c) | (b) + Spatial Guidance | 24.8549 | 0.8284 | 0.7833 | 0.1291 | 35.6026 | 1.4318 |
| (d) | (c) + Semantic Guidance | 22.7558 | 0.6652 | 0.7848 | 0.1228 | 35.4903 | 1.2442 |
| (e) | (d) + AC loss (ours) | **22.4640** | **0.6033** | **0.7849** | **0.1217** | **35.0479** | **1.2378** |

*Table 4.* Ablation study of *Interaction Fidelity* on the VVT-Interact dataset.

| ID | Method | FVD$^p$ ↓ | ISR$^p$ ↑ | FVD$^u$ ↓ | ISR$^u$ ↑ |
|---|---|---|---|---|---|
| (a) | Baseline | 419.0978 | 0.4766 | 414.3908 | 0.4763 |
| (b) | (a) + Data | 394.1350 | 0.4779 | 405.4380 | 0.4713 |
| (c) | (b) + Spatial Guidance | 384.7537 | 0.5174 | 397.2745 | 0.5110 |
| (d) | (c) + Semantic Guidance | 379.7348 | 0.5987 | 391.8533 | 0.6012 |
| (e) | (d) + AC loss (ours) | 380.5578 | **0.6100** | 393.0552 | **0.6147** |

### 4.5. Ablation Studies

Our ablation study summarized in Table 3, Table 4 and Figure 5, systematically validates that our performance stems from our novel architecture, not merely from additional training data. Critically, the results demonstrate that simply training on our VVT-Interact dataset (b) is insufficient for the interactive task. While metrics show a slight improvement over the baseline, (b) visually confirms that the model still fails to synthesize any meaningful interactions. This underscores that existing VVT architectures cannot learn complex dynamics from data alone. Furthermore, while adding Spatial Guidance (c) enables physical hand-garment contact, it cannot resolve the inherent semantic ambiguity.

The model knows where the hands are but not what they are doing. This ambiguity is effectively addressed by our Semantic Guidance (d), which provides the necessary intent. With the AC loss (e) providing further refinement, the study confirms that it is the synergistic combination of our proposed spatial and semantic guidance mechanisms that is essential for achieving high-fidelity video virtual try-on.

## 5. Limitations

While iTryOn advances interactive virtual try-on, two limitations remain. First, the model lacks explicit reasoning about garment semantics (e.g., zippers), occasionally producing "pantomimed" actions when requested to perform infeasible interactions (e.g., unzipping a seamless T-shirt). Second, while our proposed ISR metric effectively evaluates semantic success, quantifying fine-grained physical accuracy remains an open challenge for the community. We discuss these in detail in **Appendix A.1**.

## 6. Conclusion

In this work, we introduced and formalized the new task of Interactive Video Virtual Try-On (Interactive VVT). To facilitate research in this domain, we constructed the first large-scale dataset, VVT-Interact, and proposed the Interaction Success Rate (ISR) metric for standardized evaluation. To tackle the core challenges of ambiguity and sparsity, we proposed iTryOn, a framework incorporating a multi-level interaction injection mechanism and an action-aware constraint loss. Extensive experiments demonstrate that iTryOn establishes a commanding lead on the new benchmark, validating its ability to generate physically plausible interactions. Furthermore, additional evaluations on the traditional ViViD dataset confirm the model's versatility and state-of-the-art visual quality. We believe our work marks a significant step towards dynamic and immersive virtual try-on experiences.

## Acknowledgements

This work is supported by National Key Research and Development Program of China (2024YFE0203100), Scientific Research Innovation Capability Support Project for Young Faculty (No.ZYGXQNJSKYCXNLZCXM-I28), National Natural Science Foundation of China (NSFC) under Grants No.62476293 and No.62372482, and General Embodied AI Center of Sun Yat-sen University. This work was supported by Alibaba Group through Alibaba Research Intern Program.

## Impact Statement

We have carefully considered the ethical implications of our work, particularly concerning the creation of the VVT-Interact dataset and the application of our generative model. The dataset was constructed using publicly available videos from trusted sources where content creators have implicitly or explicitly consented to the public sharing of their content. To further protect personal identity, our data processing pipeline and model design are inherently privacy-preserving. The virtual try-on task is formulated to retain the head and other identifying features of the subject from the source video. The model's generative process is strictly confined to inpainting the garment and relevant limbs (hands, arms, feet), and does not reconstruct or generate facial features. This design choice mitigates the potential for misuse in creating deepfakes or otherwise compromising personal identity.

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

# A. Appendix

## A.1. Limitations and Future Work

While iTryOn marks a significant advancement in virtual try-on, we identify two key areas for future exploration.

**Handling Implausible Interactions.** Our method assumes that the input action caption corresponds to a physically feasible interaction with the target garment. However, in edge cases where the specified interaction is implausible. For example, performing an "unzipping" motion on a T-shirt that lacks a zipper, the model cannot execute the intended physical effect. In such scenarios, the framework gracefully degrades to a non-interactive virtual try-on result: it faithfully preserves the input hand motion while realistically rendering the new garment without altering its structure. The output thus resembles a plausible "pantomime" of the action, which remains visually coherent but does not reflect actual garment manipulation. This behavior highlights a current limitation: the model does not explicitly reason about garment semantics, such as the presence of zippers or buttons, when interpreting action commands. Incorporating explicit garment-aware action validation is an important direction for future work.

**Quantitative Metrics for Interaction Fidelity.** A primary challenge in the nascent field of Interactive VVT is the lack of specialized evaluation metrics. While we employ standard pixel-level (SSIM, LPIPS) and video-level (FVD, VFID) metrics, they primarily assess overall visual quality and temporal consistency rather than the specific correctness of a physical interaction. For instance, these metrics cannot distinguish between a physically plausible fabric stretch and a visually coherent but incorrect one. A crucial direction for future work is therefore the development of novel metrics designed to explicitly quantify the fidelity of human-garment interactions, potentially by analyzing fine-grained physical dynamics or semantic correctness.

## A.2. Data Annotation Pipeline

This section provides a detailed description of the annotation pipeline used to create the VVT-Interact dataset, supplementing the overview provided in Sec. 3.2.2 of the main paper. Our pipeline consists of two primary components: VLM-based annotation for semantic guidance and 3D hand prior generation.

### A.2.1. VLM-BASED ANNOTATION FOR SEMANTIC GUIDANCE

We utilized the Qwen-VL-32B model (Bai et al., 2025) for all semantic annotations due to its superior performance in our preliminary evaluations. The process was divided into caption generation and timestamp annotation.

**Caption and Interaction Type Annotation**. To generate both the global caption and the categorical action caption efficiently, we designed a single-pass inference prompt. This prompt instructs the VLM to produce a JSON object containing a high-level motion description and the specific interaction type. The predefined interaction categories are: Adjusting the collar, Adjusting the hem, Rolling/Unrolling sleeves, Putting on/Taking off clothes, Pulling at clothes, and Other interactions.

**Timestamp Annotation and Smoothing**. To acquire precise temporal boundaries for interactions, we tasked Qwen-VL-32B with a per-frame binary classification task. The raw binary labels from the VLM, however, often contain noise (e.g., isolated misclassifications). To address this, we treat the sequence of labels as a 1D signal and apply morphological operations (specifically, morphological opening followed by closing). This procedure effectively removes spurious predictions and forms coherent, continuous interaction segments, from which we extract the start and end timestamps. The detailed prompt is as follows:

> Analyze the image to determine if the person is performing a manipulative interaction with their clothing. We are only interested in purposeful actions that change or adjust the garment. Your task is to distinguish between active manipulation, passive contact, and no contact. A 'manipulative interaction' is defined as any action intended to adjust, fasten, or change the state of the garment. Consider the action 'true' only if it meets the criteria below: Pulling, tugging, or stretching the fabric to adjust its fit or position. Zipping or unzipping. Buttoning or unbuttoning. Rolling up or down sleeves. Adjusting a collar, lapel, cuff, or hemline. Putting on or taking off the garment. Actively smoothing out a wrinkle or crease with pressure. Consider the action 'false' in all other cases, especially the following: No Contact: Any pose where the hands do not touch the clothing (e.g., arms crossed, hands at sides, gesturing in the air). Passive Contact: Gently stroking or caressing the surface of the fabric without intent to adjust it. Resting Contact: Simply resting a hand or arm on the clothing without applying force to move or change it. Incidental Contact: Posing with a hand in a pocket, where the primary action isn't adjusting the pocket itself. Based on these detailed definitions, is a manipulative interaction occurring in the image? Please respond with only the word 'true' or 'false'.

### A.2.2. VLM MODEL SELECTION

To select the optimal VLM for our annotation pipeline, we conducted a comparative study between the Qwen-VL (Bai et al., 2025) and Gemma3 series (Team et al., 2025), chosen for their strong performance and efficiency. We manually annotated a test set of 1,000 frames for the binary interaction classification task and evaluated each model's performance. The results are summarized in Table 5. As shown, Qwen-VL-32B achieves the highest F1-score and precision. We note that this annotation

*Table 5.* Comparison of different VLMs for the per-frame interaction annotation task. Qwen-VL-32B demonstrates the best overall performance, particularly in F1-score and precision.

| Model | Accuracy | Precision | Recall | F1-score |
|---|---|---|---|---|
| Qwen-VL 7B | 62.8 | 60.0 | 75.0 | 66.6 |
| **Qwen-VL 32B** | **74.1** | **79.7** | 69.8 | **74.4** |
| Gemma3 12B | 57.3 | 54.9 | 77.6 | 64.3 |
| Gemma3 27B | 56.9 | 54.0 | **87.2** | 66.7 |

task has inherent ambiguity, particularly in identifying the exact start and end frames of an interaction. Given this context, the performance of Qwen-VL-32B is considered highly effective for our large-scale automated annotation requirements.

### A.3. 3D Hand Prior Annotation

The 3D hand prior is generated using HaMeR (Pavlakos et al., 2024), which we applied on a per-frame basis to estimate the 3D hand mesh and pose from the input video. The resulting 3D information was then rendered into 2D image representations to be used as spatial guidance. A manual inspection of a random subset of the data revealed a high accuracy rate, exceeding 95%. Furthermore, the overall framework is robust to minor inaccuracies in the 3D hand prior, as the DWpose (Yang et al., 2023) features provide a foundational and reliable representation of the overall body and hand position.

### A.4. Further Details on Action-aware Semantic Guidance

This section provides a deeper analysis of our Action-aware Semantic Guidance module. We first present qualitative examples to motivate the need for explicit action captions, and then provide detailed quantitative ablation studies to validate the effectiveness of each component.

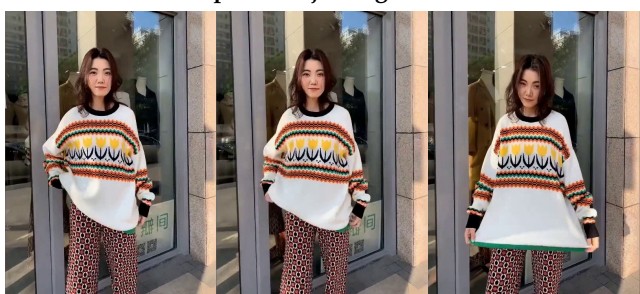

**Example 1: Rolling Sleeves**

**Global caption**: The person stands still, **adjusts** shirt sleeves and collar, moves hands, and shifts body slightly.
**Our Action caption**: Rolling/Unrolling sleeves

**Example 2: Adjusting the Hem**

**Global caption**: The person stands still, turns slightly, and **adjusts** the sweater.
**Our Action caption**: Adjusting the hem

*Figure 6.* Visual motivation for our Action-aware Semantic Guidance. These examples from our VVT-Interact dataset highlight the semantic ambiguity of VLM-generated global captions. Although the ground-truth interactions are distinct (rolling sleeves vs. adjusting the hem), both are imprecisely described with the generic verb "adjusts". Our categorical action captions resolve this ambiguity, providing the model with a clear and actionable signal required for high-fidelity interaction synthesis.

### A.4.1. MOTIVATION: AMBIGUITY IN GLOBAL CAPTIONS

As stated in the main paper, a key motivation for our work is the inherent ambiguity of high-level motion descriptions generated by VLMs. While these global captions provide a useful summary, they often fail to capture the specific nature of a

human-garment interaction, using generic verbs for distinct actions. This semantic ambiguity acts as a confusing supervisory signal, causing the model to default to the easier task of generating a non-interactive try-on rather than attempting a specific complex interaction. Figure 6 presents concrete examples from our VVT-Interact dataset that illustrate this problem.

### A.4.2. QUANTITATIVE ABLATION STUDY

To validate the contribution of our proposed components, we conduct a detailed ablation study. As discussed in the main paper, the transition from model (c) to (d) in Table 3 highlights the impact of our full Semantic Guidance module. We dissect this gain by incrementally adding the action caption and A-RoPE to the baseline with spatial guidance (c). The results are presented in Table 6.

- Benefit of Action Captions: Comparing model (c) and (d'), we observe a consistent improvement across most metrics after introducing the time-stamped action captions. This confirms that providing the model with an explicit semantic signal about the action's type is crucial for resolving the ambiguity demonstrated above and improving generation quality.

- Crucial Role of A-RoPE: The subsequent addition of A-RoPE in model (d) yields another performance leap. The improvement is particularly pronounced in metrics sensitive to temporal consistency. This validates our hypothesis that precisely synchronizing the textual guidance with the corresponding video frames is critical. A-RoPE prevents the semantic information from "bleeding" into non-interactive frames and empowers the model to generate actions with accurate timing.

*Table 6.* Detailed ablation study on the components of Semantic Guidance.

| ID | Method | $\text{VFID}_I^p \downarrow$ | $\text{VFID}_R^p \downarrow$ | SSIM↑ | LPIPS↓ | $\text{VFID}_I^u \downarrow$ | $\text{VFID}_R^u \downarrow$ |
|---|---|---|---|---|---|---|---|
| (c) | Baseline + Data + Spatial Guidance | 24.8549 | 0.8284 | 0.7833 | 0.1291 | 35.6026 | 1.4318 |
| (d') | (c) + Action caption | 23.5618 | **0.5584** | 0.7823 | 0.1232 | 36.0608 | 1.2831 |
| (d) | (c) + Action caption + A-RoPE | **22.7558** | 0.6652 | **0.7848** | **0.1228** | **35.4903** | **1.2442** |

A key hyperparameter in our A-RoPE design is the separation scale $k$ from Equation 2. This parameter controls how distinctly different action segments are encoded in the positional space. To determine the optimal value, we performed an ablation study on $k$, with results shown in Table 7.

*Table 7.* Ablation study on the separation scale hyperparameter $k$ in A-RoPE. The value $k = 4$ yields the best overall performance.

| Method | $\text{VFID}_I^p \downarrow$ | $\text{VFID}_R^p \downarrow$ | SSIM↑ | LPIPS↓ | $\text{VFID}_I^u \downarrow$ | $\text{VFID}_R^u \downarrow$ |
|---|---|---|---|---|---|---|
| $k=2$ | 23.8789 | **0.6124** | 0.7825 | 0.1247 | 35.6738 | 1.7365 |
| $k=4$ | **22.7558** | 0.6652 | **0.7848** | **0.1228** | **35.4903** | **1.2442** |
| $k=6$ | 22.8378 | 0.6852 | 0.7844 | 0.1341 | 35.5785 | 1.2652 |

In conclusion, this detailed analysis validates our approach to Action-aware Semantic Guidance. We have first demonstrated that categorical action captions are essential for resolving the critical semantic ambiguity found in global prompts, which can cause the model to default to simpler non-interactive generations. Subsequently, we have shown that our proposed A-RoPE mechanism is crucial for enforcing the temporal precision required to synchronize this powerful guidance. The synergistic combination of these two components is key to empowering the model to generate accurate interactions.

### A.5. Ablation Study of Action Constraint Loss Weight

A key hyperparameter in our action-aware constraint loss (AC loss) is the weighting coefficient $\lambda$ in Equation 3. This parameter governs the strength of the additional supervision applied to interactive frames, as defined by the action mask $\mathbb{M}_{\text{action}}$. A higher $\lambda$ places greater emphasis on accurately reconstructing interaction segments during diffusion training, thereby counteracting the dominance of non-interactive frames. To identify the optimal trade-off, we conduct an ablation study over $\lambda$, with results reported in Table 8.

*Table 8.* Ablation study on the weighting coefficient $\lambda$ in AC loss. The value $\lambda = 0.5$ yields the best overall performance.

| Method | VFID$_I^p$ ↓ | VFID$_R^p$ ↓ | SSIM↑ | LPIPS↓ | VFID$_I^u$ ↓ | VFID$_R^u$ ↓ |
|---|---|---|---|---|---|---|
| $\lambda$=0.0 | 22.7558 | 0.6652 | 0.7848 | 0.1228 | 35.4903 | 1.2442 |
| $\lambda$=0.5 | **22.4640** | 0.6033 | **0.7849** | **0.1217** | 35.0479 | **1.2378** |
| $\lambda$=1.0 | 23.4714 | **0.5814** | 0.7833 | 0.1242 | 35.5598 | 1.2612 |
| $\lambda$=2.0 | 24.3152 | 1.9134 | 0.7837 | 0.1309 | **34.8644** | 2.5655 |

## A.6. Additional Experiments and Analysis on Non-Interactive VVT

Although the primary contribution of iTryOn lies in the new Interactive VVT task, we recognize the importance of validating our framework on established standards. In this section, we first present our state-of-the-art performance on the widely-used non-interactive ViViD benchmark. Subsequently, we provide a detailed analysis to clarify that this superior performance stems from our strategic choice of a foundational backbone and advanced general-purpose training strategies, rather than our interaction-specific innovations.

### A.6.1. PERFORMANCE ON THE VIVID BENCHMARK

We benchmark iTryOn against leading methods including ViViD (Fang et al., 2024), CatV$^2$TON (Chong et al., 2025b), MagicTryOn (Li et al., 2025b), and DreamVVT (Zuo et al., 2025) on the standard ViViD-S-Test (Chong et al., 2025b).

**Quantitative Comparison.** As reported in Table 9, iTryOn achieves a decisive lead across almost all metrics. Notably, we surpass MagicTryOn in VFID and SSIM, despite our model being significantly more parameter-efficient. This demonstrates that iTryOn produces videos with higher visual fidelity and better temporal consistency.

**Qualitative Comparison.** Figure 7 provides visual comparisons. iTryOn excels in preserving intricate garment details and maintaining structural integrity during motion, whereas baseline methods often exhibit blurring or temporal flickering.

*Table 9.* Quantitative comparison on the non-interactive ViViD dataset. iTryOn achieves state-of-the-art results, outperforming models with significantly larger parameter counts.

| Method | Params | VFID$_I^p$ ↓ | VFID$_R^p$ ↓ | SSIM↑ | LPIPS↓ | VFID$_I^u$ ↓ | VFID$_R^u$ ↓ |
|---|---|---|---|---|---|---|---|
| ViViD (Fang et al., 2024) | 2B | 17.2924 | 0.6209 | 0.8029 | 0.1221 | 21.8032 | 0.8212 |
| CatV$^2$TON (Chong et al., 2025b) | 5B | 13.5962 | 0.2963 | 0.8727 | 0.0639 | 19.5131 | 0.5283 |
| MagicTryOn (Li et al., 2025b) | 14B | 12.1988 | 0.2346 | 0.8841 | 0.0815 | 17.5710 | 0.5073 |
| DreamVVT (Zuo et al., 2025) | - | 11.0180 | 0.2549 | 0.8737 | **0.0619** | 16.9468 | 0.4285 |
| **iTryOn (ours)** | 2B | **8.4322** | **0.0876** | **0.8944** | 0.0679 | **13.2806** | **0.2293** |

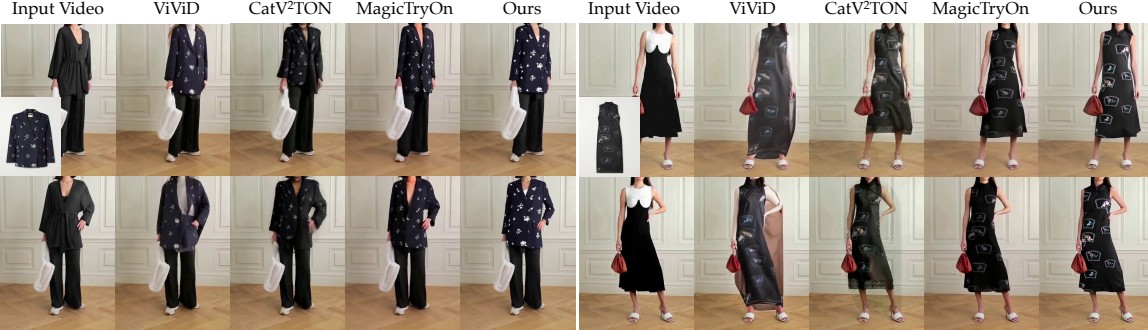

*Figure 7.* Qualitative comparison on the ViViD dataset.

## A.6.2. ANALYSIS OF PERFORMANCE DRIVERS

The strong performance on ViViD might seem surprising given our focus on interactive scenarios. Here, we clarify that this success is not an anomaly but the result of two key factors: a foundational backbone inherently suited for VVT and the application of advanced general-purpose training strategies.

**A Foundational Backbone Inherently Suited for VVT.** While contemporary methods like MagicTryOn (Li et al., 2025b) and DreamVVT (Zuo et al., 2025) are also built on powerful video generation models, our choice of Wan2.1-VACE (Jiang et al., 2025) offers a distinct advantage. Wan2.1-VACE is pre-trained for reference-guided editing, which aligns perfectly with the VVT task definition: video inpainting conditioned on a high-fidelity reference image (garment) and structural control (pose). By inheriting the strong priors for preserving textural identity from Wan2.1-VACE, our framework gains a "head start" in maintaining garment fidelity and temporal coherence, forming a powerful baseline even before specific interaction modules are added.

**Advanced Training and Inference Strategies.** Beyond the backbone, we incorporate general-purpose techniques to enhance efficiency and quality. We conduct an ablation study starting from the Wan2.1-VACE backbone fine-tuned on ViViD, incrementally adding two strategies (Table 10):

- **Loss Weighting:** We apply a flow-matching loss weighting scheme[1] to accelerate convergence and stabilize fine-tuning. Comparing row (1) and (2) in Table 10, this yields consistent gains.

- **Interval Guidance:** During inference, we employ Interval Guidance (Kynkäänniemi et al., 2024) to apply Classifier-Free Guidance (CFG) only during the early sampling steps (e.g., first 10%-40%). This prevents oversaturation and artifacts common in full-process CFG. The transition from (2) to (3) highlights the substantial benefit of this technique.

This analysis confirms that our SOTA performance on ViViD is driven by these general-purpose strengths, which provide a robust foundation for our interaction-specific innovations to build upon.

*Table 10.* Ablation of general-purpose enhancements on the ViViD dataset. These strategies significantly boost performance independent of interaction modules.

| ID | Method | $\text{VFID}_I^p \downarrow$ | $\text{VFID}_R^p \downarrow$ | SSIM↑ | LPIPS↓ | $\text{VFID}_I^u \downarrow$ | $\text{VFID}_R^u \downarrow$ |
|---|---|---|---|---|---|---|---|
| (1) | Wan2.1-VACE (SFT) (Jiang et al., 2025) | 9.6956 | 0.1367 | 0.8892 | 0.0704 | 14.1810 | 0.4060 |
| (2) | (1) + Loss Weight | 9.3765 | 0.1308 | 0.8894 | 0.0703 | 13.6870 | **0.1890** |
| (3) | (2) + Interval Guidance | **8.4697** | **0.0831** | **0.8948** | **0.0663** | **13.3776** | 0.1911 |

---

[1] https://github.com/modelscope/DiffSynth-Studio/blob/main/diffsynth/diffusion/flow_match.py

