# OpenReview forum: "iTryOn: Mastering Interactive Video Virtual Try-On with Spatial-Semantic Guidance"
_ICML.cc/2026/Conference — ICML 2026 regular_

### Official Review · Reviewer_YQnP · 2026-03-02

**Soundness:** 3
**Presentation:** 4
**Significance:** 4
**Originality:** 3
**Overall Recommendation:** 4
**Confidence:** 4

**Summary:**

This paper introduces the novel task of Interactive Video Virtual Try-On. Authors argue that existing virtual try-on benchmarks and methods primarily focus on passive poses and struggle with realistic scenarios where subjects actively manipulate their clothing. To address this, the paper proposes a framework that The method introduces two key conditioning mechanisms: (1) spatial guidance using a garment-agnostic 3D hand prior to resolve contact ambiguity, and (2) a temporal-semantic guidance via time-stamped action captions synchronized using an Action-aware RoPE mechanism. Additionally, the authors propose an action-aware constraint loss to emphasize interactive frames during training. To support this, the paper introduces the VVT-Interact dataset and a new VLM-based evaluation metric (ISR).

**Compliance With Llm Reviewing Policy:**

Affirmed.

**Final Justification:**

The rebuttal adequately addresses my main concerns. In particular, the authors directly tackle the ISR circularity issue by re-evaluating all methods with an independent evaluator and showing that the relative ranking is unchanged, which substantially increases my confidence that the reported gains are not simply due to Qwen-VL-specific bias. They also clarify the baseline-fairness issue. The explanation of the AC-loss sensitivity is also reasonable, and the deployment-time timestamp extraction pipeline is now clear. My remaining concerns are mostly about longer-term scalability to open-vocabulary interactions, which I view as a scope limitation rather than a rebuttal-level blocker.

**Key Questions For Authors:**

1) Were ViViD, CatV2TON, and MagicTryOn fine-tuned on the VVT-Interact dataset? If so, what were their conditioning inputs and training budgets? If not, can you provide results for at least one strong baseline that has been fairly adapted to VVT-Interact?

2) Can you validate the ISR metric by providing either (i) a small human study showing correlation with human judgments, or (ii) results using an evaluator VLM completely independent from the one used for annotation?

3) Why does a higher lambda degrade visual fidelity metrics so sharply? Do your training curves or qualitative failure patterns explain this instability?

4) How do you plan obtaining time-stamped action captions during real-world inference?

**Limitations:**

The authors adequately note limitations regarding infeasible "pantomimed" actions and the lack of fine-grained physical metrics. However, they should expand this section to explicitly discuss the generalization limits imposed by a closed action taxonomy. Furthermore, they must acknowledge the inherent risks of relying on a single VLM in both the annotation and evaluation loops for the ISR metric.

**Strengths And Weaknesses:**

Strengths:
- Formalizing Interactive VVT is a highly practical and meaningful step forward for the field. Releasing a dedicated dataset and a targeted interaction metric is likely to spur valuable follow-up work in e-commerce and live-stream try-on applications.
- The motivation behind addressing interaction ambiguity and sparse interactive frames is clear. Furthermore, the architectural choices—such as opting for a 3D-hand prior to avoid the geometry leakage seen in depth maps, and the specific timing of captions—are clearly explained and beautifully illustrated.
- The technical design is logically coherent and largely supported by strong ablation studies. Authors successfully isolate the individual performance contributions of the spatial guidance, the A-RoPE semantic guidance, and the AC loss.
- While built upon a strong existing backbone, the combination of garment-agnostic 3D hand priors with segment-synchronized semantic conditioning is a highly creative, task-tailored extension that moves well beyond standard VVT conditioning techniques.

Weaknesses:
- The paper uses Qwen-VL to generate the interaction timestamps and action captions during dataset annotation, and then uses that same model to verify interaction success when computing the ISR metric. This might create a risk of circularity bias, since the metric may simply be rewarding the evaluator's own inductive biases rather than reflecting true, human-perceived interaction fidelity.
- The experimental protocol lacks clarity regarding baseline fairness. iTryOn is trained in two stages (first on ViViD, then fine-tuned on VVT-Interact), but it is not explicitly stated whether the competing baselines were given the same fine-tuning budget on VVT-Interact or what conditioning inputs they received. Thus, it is difficult to attribute the empirical gains purely to the proposed architecture.
- The framework currently categorizes interactions into six predefined types. The paper would benefit from a brief discussion on how this system might scale to open-vocabulary interactions or handle entirely unseen manipulation types in the wild.
- The reported hyperparameter sweep reveals that the AC loss weight is quite critical, with performance degrading sharply at higher values. A brief diagnostic explaining this instability would improve confidence in the model's robustness.

---

> ### Author Rebuttal · Authors · 2026-03-31
>
> **w1 & q2:** Circularity Bias in ISR Metric (Validation with Independent VLM)
>
> This is a valid concern. To prove that our performance gains are not exploiting Qwen-VL's inductive biases, we re-evaluated all models using an entirely independent evaluator: Gemini Flash. As shown in the table below, while the absolute scores differ slightly from Qwen-VL, the relative performance rankings remain identical, with iTryOn maintaining a commanding lead. Furthermore, Qwen-VL achieves a 74.1% consistency rate with human annotators (Appendix A.2.2).
> | Method | ISR (Paired) | ISR (Unpaired) |
> | :--- | :---: | :---: |
> | ViViD | 0.3846 | 0.3571 |
> | CatV2TON | 0.4545 | 0.4342 |
> | MagicTryOn | 0.4246 | 0.4363 |
> | **iTryOn (Ours)** | **0.6375** | **0.6515** |
>
> **w2 & q1:** Baseline Fairness / Fine-tuning
>
> Because open-source VVT models only provide inference code, we could not fairly fine-tune them on VVT-Interact without risking suboptimal training configurations. To provide a rigorous baseline comparison, we rely on our "(a) + Data" ablation study (Tables 3 & 4). This acts as our "fairly adapted baseline": it is our strong DiT backbone fine-tuned on the VVT-Interact dataset. The results clearly show that simply adding interactive data without our proposed spatial/semantic guidance fails to synthesize interactions.
>
> **w3:** Scaling to Open-Vocabulary Interactions / Unseen Manipulations
>
> Regarding the handling of unseen interactions in the wild, our current dataset intentionally includes an "Other interactions" category, which equips the model with a certain degree of generalization for less common manipulations. Furthermore, our framework already possesses a foundation for open-vocabulary understanding, as the injected global caption leverages continuous semantic representations to capture diverse contexts beyond the predefined categories. If the framework encounters a truly Out-Of-Distribution interaction that it still cannot physically execute, it does not catastrophically fail. Instead, it gracefully degrades to a standard non-interactive virtual try-on: it faithfully preserves the user's hand motion while realistically rendering the garment without physical deformation (as discussed in Appendix A.1). Ultimately, to fully scale to open-vocabulary interactions, the fundamental bottleneck is data. Our future work will focus on significantly expanding the dataset to cover a much wider, unconstrained variety of physical manipulations in the wild.
>
> **q3:** Instability of AC Loss at Higher Lambda
>
> The sharp degradation at higher lambda values stems from the severe sparsity of interactive frames. In a typical 81-frame clip, only about ~30 frames contain interactions. If lambda is set too high, the model's gradients become overwhelmingly dominated by this minority of frames, leading to a severe training imbalance. While the training loss curves may not show obvious divergence, the inference results visually manifest this imbalance as localized artifacts, overexposure, and temporal flickering in non-interactive frames. lambda=0.5 strikes the necessary balance.
>
> **q4:** Obtaining Timestamps in Real-World Inference
>
> Because the VVT task requires a source video as input (to extract pose/motion), we use the exact same automated VLM pipeline during real-world inference. When a user uploads a video, we run it through our pre-processing pipeline to automatically extract the time-stamped action captions and 3D hand priors before passing them to the model.

---

> > ### Author Rebuttal · Reviewer_YQnP · 2026-04-01
> >
> > The rebuttal properly addresses my main concerns by adding an independent evaluator for ISR and a convincing data-only adapted baseline argument. My remaining concerns are mostly about longer-term scalability to open-vocabulary interactions, which I view as a scope limitation rather than a rebuttal-level blocker.

---

### Official Review · Reviewer_yPPQ · 2026-03-08

**Soundness:** 3
**Presentation:** 4
**Significance:** 2
**Originality:** 3
**Overall Recommendation:** 3
**Confidence:** 4

**Summary:**

This paper explores the domain of Interactive Video Virtual Try-On, extending the traditional task of passive garment display to dynamic scenarios where subjects actively manipulate their clothing. To overcome the semantic ambiguity of standard pose estimation and the challenge of learning from sparse interactive moments, the authors propose the iTryOn framework, built upon a video diffusion Transformer. iTryOn introduces a multi-level interaction injection mechanism: at the spatial level, it utilizes a garment-agnostic 3D hand prior to guide precise physical contact; at the semantic level, it integrates global and time-stamped action captions via an Action-aware Rotational Position Embedding for precise temporal alignment. Furthermore, an action-aware constraint loss is introduced to focus supervision on critical interaction frames. To support this new direction, the authors construct a new dataset, VVT-Interact, and propose a VLM-based Interaction Success Rate metric, with experiments demonstrating strong performance on both interactive and non-interactive benchmarks.

**Compliance With Llm Reviewing Policy:**

Affirmed.

**Final Justification:**

I thank the authors for the detailed rebuttal and the additional experiments.
However, my main concerns remain only partially addressed. The ISR metric still shows a non-negligible gap with human judgments. In addition, the method’s generalization ability remains constrained, as it does not handle more complex scenarios such as multi-person interactions or severe occlusions, which limits its broader applicability.
While the rebuttal strengthens the paper, it does not sufficiently change my assessment regarding evaluation robustness and generalization. Therefore, I maintain my weak reject recommendation.

**Key Questions For Authors:**

1. Have you evaluated the consistency between the ISR metric (which relies on Qwen-VL) and human annotations? If the VLM makes incorrect judgments, how does this impact the overall evaluation results and their reliability?
2. Can the proposed method stably generate human-garment interactions in more complex scenarios, such as videos featuring multiple people, severe occlusions, or highly cluttered backgrounds?
3. Could you provide a deeper analysis of the performance differences across specific interaction types (e.g., zipping vs. pulling the hem)?

**Limitations:**

yes

**Strengths And Weaknesses:**

**Strengths:**
1. The paper introduces a new task definition. Traditional VVT methods primarily focus on passive display, whereas this paper formalizes Interactive VVT to capture active human-garment interactions. This has significant real-world applicability for e-commerce and live-streaming.
2. The proposed framework is systematic. iTryOn injects interaction information at two levels: spatially via a 3D hand prior to resolve the depth and orientation ambiguity of 2D poses, and semantically through time-stamped captions and A-ROPE to synchronize text prompts with specific video segments. Coupled with the action-aware constraint loss.
3. The construction of the VVT-Interact dataset, comprising 5,292 video-garment pairs , alongside the introduction of the VLM-based ISR metric, lays a foundation for future research and evaluation in this sub-field.


**Weaknesses:**
1. The baseline comparisons are somewhat limited. The experiments primarily compare iTryOn against ViViD, CatV2TON, and MagicTryOn. Given the rapid advancements in video generation and editing, the current scope is relatively narrow. Including qualitative visual comparisons with other recent diffusion-based VVT or video editing models in the supplementary materials would better demonstrate the method's overall superiority.

2. The reliability and objectivity of the ISR metric are questionable. The metric relies entirely on a Qwen-VL to automatically verify the generated actions. This evaluation process could introduce the VLM's own biases or "hallucinations". The paper lacks an analysis of ISR's consistency with human evaluation and acknowledges that the metric cannot quantify fine-grained physical accuracy.

3. The framework heavily relies on pre-trained priors. iTryOn depends significantly on external models, such as the Wan2.1-VACE backbone , HaMeR for 3D hand reconstruction , and Qwen-VL for annotation. The performance on the non-interactive ViViD benchmark is partially attributed to the inherent advantages of the Wan2.1-VACE backbone rather than interaction-specific modules.

4. The model fails to handle implausible interactions. The framework lacks explicit semantic reasoning about garment structures. Consequently, when prompted to perform an infeasible action (e.g., unzipping a seamless T-shirt), it generates a physically implausible "pantomimed" action without actively engaging the garment.

5. There is no discussion of computational overhead. Introducing 3D hand feature extraction and parallel context blocks  inevitably increases computational load. The paper omits quantitative comparisons of inference latency, VRAM usage, or FLOPs against baselines, making it difficult to assess real-world deployment costs.

---

> ### Author Rebuttal · Authors · 2026-03-31
>
> **w1:** Baselines and Video Editing Models
>
> We selected ViViD, CatV2TON, and MagicTryOn as they are the current state-of-the-art in open-source video virtual try-on. General video editing models typically perform poorly on garment-preserving try-on tasks without extensive domain-specific fine-tuning. However, we agree that adding qualitative comparisons to general editing models will strengthen our paper, and we will include these in the final version.
>
> **w2 & q1:** Reliability of ISR Metric & Consistency with Human Evaluation
>
> As detailed in Appendix A.2.2, we conducted a human evaluation to validate the VLM's judgment. The Qwen-VL 32B model achieves a 74.1% consistency rate with human annotators on the binary interaction classification task. Given that identifying the exact start/end frames of an interaction contains inherent subjective ambiguity (due to motion blur or self-occlusion), this 74.1% agreement is highly competitive. For context, in the image-reward domain, current SOTA models align with single-human judgments at around 77%. Furthermore, since the same VLM is used to evaluate all models, the relative ranking remains objective and fair.
>
> **w3:** Reliance on Pre-trained Priors & Backbone
>
> While we utilize Wan2.1-VACE as our backbone, it is common practice in VVT to leverage advanced foundation models (e.g., MagicTryOn utilizes a 14B backbone but still underperforms our 2B model). Our state-of-the-art results on the non-interactive ViViD benchmark simply serve to prove that our interaction-specific modules do not degrade standard VVT performance. The core contribution remains how we successfully inject sparse interaction conditions into this backbone.
>
> **w4:** Implausible Interactions
>
> We acknowledge this limitation. In practical deployments, a straightforward and effective solution is to introduce a VLM-based front-end filter to validate the prompt against the source video/garment. If a user requests an impossible action (e.g., "unzip a seamless T-shirt"), the system will reject or rewrite the prompt before generation.
>
> **w5:** Computational Overhead
>
> Since the parallel Context Blocks are also present in the baseline architecture, the only additional overhead introduced by our method is the Interaction Guider, which consists of just one lightweight block. As shown below, iTryOn is significantly more efficient than the current SOTA MagicTryOn:
> | Method | Parameters | Inference Time (per video) | VRAM Usage |
> | :--- | :--- | :--- | :--- |
> | MagicTryOn | 14B | 461 sec | 61.9 GB |
> | **iTryOn (Ours)** | **2B** | **187 sec** | **38.4 GB** |
>
> **q2:** Complex Scenarios (Multiple People, Occlusion, Cluttered Backgrounds)
>
> Current VVT datasets do not yet cover multi-person interactions. Extending this to multiple people and severe occlusions is an important direction for future work. However, our model can successfully handle highly cluttered backgrounds, as the background is treated as a protected region.
>
> **q3:** Performance Differences Across Interaction Types
>
> The performance discrepancy (e.g., zipping vs. pulling the hem) is primarily due to the real-world data distribution. In our dataset, "Adjusting the hem" and "Adjusting the collar" account for ~74% of the data, while "Zipping" falls under the "Other interactions" category (<3%). Consequently, human evaluations show a success rate of ~80% for hem-pulling but only ~20% for zipping. Addressing this long-tail data imbalance in interactive generation is a priority for our future work.

---

> > ### Author Rebuttal · Reviewer_yPPQ · 2026-04-02
> >
> > The rebuttal addresses some of my concerns, but it is not enough to change my overall recommendation. Therefore, I will keep my score at 3.
> >
> > The authors provided helpful clarifications on computational costs and admitted the limitations regarding implausible interactions. The explanation of performance across different interaction types also makes the "long-tail" challenge clearer. However, my main concerns are still not fully fixed:
> >
> > * **ISR Metric:** A 74.1% consistency with humans suggests this metric is not fully reliable. Since ISR is a central part of the evaluation, the paper still needs a deeper analysis of how VLM errors might change the results.
> > * **Limited Baselines:** Promising to add more comparisons in the final version does not solve the current problem. Right now, the empirical comparison is still too narrow.
> > * **Generalization:** The rebuttal confirms that the method cannot handle complex scenes like multi-person interactions or heavy occlusions. This limits the usefulness of the approach.
> >
> > The rebuttal makes the paper clearer, but it does not sufficiently solve the key issues regarding evaluation and generalization.

---

> > > ### Author Response · Authors · 2026-04-02
> > >
> > > We sincerely thank you for acknowledging our rebuttal and engaging in this constructive discussion. We completely understand your remaining concerns. To address them decisively during this review phase, we have conducted additional experiments to provide a better empirical comparison.
> > >
> > > ### Baseline Comparisons (General Video Editing Models)
> > >
> > > We have immediately evaluated a state-of-the-art general video editing model, **Wan2.2-Animate-14B**, directly on our VVT-Interact test set. We selected this model because it represents the cutting-edge of general video editing, supports reference image inputs, and excels at human animation.
> > >
> > > However, as we hypothesized, general video editing models lack the garment-specific structural priors required for virtual try-on. Below are the quantitative results (Unpaired setting):
> > >
> > > | Method | FVD (unpair) ↓ | ISR (unpair) ↑ |
> > > | :--- | :--- | :--- |
> > > | ViViD | 482.2153 | 0.3888 |
> > > | CatV2TON | 542.4718 | 0.4245 |
> > > | MagicTryOn | 432.3735 | 0.4474 |
> > > | Wan2.2-Animate-14B | 471.1892 | 0.4912 |
> > > | **iTryOn (Ours)** | **393.0552** | **0.6147** |
> > >
> > > Despite having significantly more parameters, Wan2.2-Animate-14B struggles to match the temporal coherence (FVD) and interaction fidelity (ISR) of our 2B iTryOn framework. To visually demonstrate why this happens, we have updated our qualitative comparison videos to include the Wan2.2-Animate-14B results. You can download and view the video from our anonymous GitHub repository here:
> > > 🔗 **https://github.com/itryon/itryon_supp/blob/main/additional_supp.zip**
> > >
> > > As the video clearly shows, without domain-specific VVT modules, the general model struggles severely with garment texture preservation, leading to significant structural collapse. This empirical comparison directly confirms our assertion: general video editing models are not readily competitive for the strict garment-preserving requirements of VVT.
> > >
> > > ### Deeper Analysis of the ISR Metric & VLM Errors
> > >
> > > To directly answer your question about *how VLM errors might change the evaluation results*: **they do not alter the relative model rankings.** To prove that our evaluation is not skewed by Qwen-VL's specific biases, we re-evaluated all generated videos using a different evaluator: Gemini Flash.
> > >
> > > The results show that while absolute scores vary slightly, the performance gap and relative rankings remain exactly the same, with iTryOn maintaining a commanding lead. This confirms that inherent VLM errors are uniformly distributed and do not unfairly favor our method.
> > >
> > > | Method | ISR (Paired) | ISR (Unpaired) |
> > > | :--- | :--- | :--- |
> > > | ViViD | 0.3846 | 0.3571 |
> > > | CatV2TON | 0.4545 | 0.4342 |
> > > | MagicTryOn | 0.4246 | 0.4363 |
> > > | **iTryOn (Ours)** | **0.6375** | **0.6515** |
> > >
> > > ### Generalization vs. Scope of the Paper
> > >
> > > We respectfully request the reviewer to consider the fundamental scope of our work. We claim to explore the domain of *Interactive* Video Virtual Try-On, which is a completely new frontier. Handling multi-person interactions and severe occlusions remains a highly challenging unsolved problem even in traditional non-interactive video try-on.
> > >
> > > Overall, this article's broad aspect comprises formalizing this new interactive task, constructing the very first dataset, and proposing a framework that successfully bridges the gap from "non-interactive" to "interactive".
> > >
> > > We hope the addition of the Wan2.2-Animate baseline, the visual comparisons, and the Gemini evaluation fully addresses your remaining reservations. We kindly ask you to reconsider your score in light of these new results. Thank you again for your time and rigorous review!

---

### Official Review · Reviewer_iRjV · 2026-03-13

**Soundness:** 3
**Presentation:** 3
**Significance:** 3
**Originality:** 3
**Overall Recommendation:** 4
**Confidence:** 3

**Summary:**

This paper proposes a new task, Interactive VVT, which tackles video virtual try-on for motions involving human–garment interactions. To support this task, the paper introduces a dataset (VVT-Interact) curated from e-commerce live streams and social media. Built on this benchmark, it proposes iTryOn, a framework for Interactive VVT based on a video diffusion transformer. Specifically, the method introduces a multi-level interaction injection mechanism and an action-aware constraint loss. Experiments show that the proposed method outperforms existing baselines on the collected VVT-Interact dataset.

**Compliance With Llm Reviewing Policy:**

Affirmed.

**Final Justification:**

I find the authors’ response reasonable and will maintain my positive rating.

**Key Questions For Authors:**

See the Weaknesses section above.

**Limitations:**

Yes.

**Strengths And Weaknesses:**

**Strengths**

**(1) Good writing quality.**

The paper is easy to read, and the proposed task and method are well explained with reasonable motivations.

**(2) Novel task.**

The proposed problem formulation is novel, and the curated dataset for this task could be valuable to the research community.

**(3) Strong experimental results.**

The proposed model clearly outperforms the baselines in both interactive and non-interactive video virtual try-on.

**Weaknesses**

**(1) Questionable robustness to preprocessing models.**

The proposed framework heavily depends on the outputs of several preprocessing models, e.g., 3D hand reconstruction from HaMeR and captions generated by Qwen-VL. It is unclear how robust the method is to errors in these estimated inputs.

**(2) Questionable comparison setting.**

While the experiments show that the proposed model clearly outperforms existing baselines on the curated VVT-Interact dataset, it is unclear whether this gain comes from the framework design itself or from the fact that it is the only model fine-tuned on the VVT-Interact training set while the other models are not.

**(3) Lack of diverse video results.**

Although this is a video generation task, the paper reports only two video results per comparison setting. It would be helpful to include more video results in a revision.

---

> ### Author Rebuttal · Authors · 2026-03-31
>
> **Regarding Weaknesses:**
>
> **w1**: Robustness to preprocessing models (3D hand & VLM captions)
>
> We agree that reliance on preprocessing models introduces potential noise. However, our framework is designed to be highly robust to minor inaccuracies in these inputs. As discussed in Appendix A.3, the DWPose features provide a foundational, reliable representation of the overall body and hand positions, acting as a structural safety net. Furthermore, our Action-aware Semantic Guidance complements the global captions. Even if the VLM or HaMeR outputs contain minor errors, the synergistic combination of pose priors and multi-level semantic guidance ensures the model gracefully falls back to generating plausible try-on results without catastrophic failure.
>
> **w2**: Questionable comparison setting / Fairness against baselines
>
> The primary reason we did not fine-tune the baselines (ViViD, CatV2TON, MagicTryOn) on VVT-Interact is that their official repositories currently only provide inference code, making it difficult to reproduce their exact training settings. Retraining them with guessed configurations would likely yield suboptimal results, leading to an unfair comparison.
> To rigorously address this, we conducted an ablation study labeled "(a) + Data" in Tables 3 and 4. This setting represents our DiT baseline fine-tuned on the VVT-Interact dataset without our proposed interaction modules. As shown in the tables and the supplementary videos, merely fine-tuning on interactive data fails to yield meaningful interactions. This isolates the source of our performance gain, proving that the improvement stems directly from our novel framework design (spatial/semantic guidance and AC loss), not just the new training data.
>
> **w3**: Lack of diverse video results
>
> Thank you for the suggestion. In the final version, we will provide much more comprehensive supplementary video showcases across all interaction categories.

---

> > ### Author Rebuttal · Reviewer_iRjV · 2026-04-04
> >
> > I appreciate the authors’ discussion and find it reasonable. Regarding the limitations additionally discussed in the rebuttal, I strongly encourage the authors to include them in the revision.

---

### Decision · Program_Chairs · 2026-04-30

**Decision:**

Accept (regular)

**Comment:**

This paper received mixed positive and negative scores.
Although `Reviewer yPPQ` is still negative to this paper, the raised concern on generalization to multi-person and occlusion cases are out of this work's scope.
The authors clarified all other concerns to all reviewers.
Hence, AC recommends accepting this work.
It is strongly recommended to include valuable points raised in the rebuttal period.